# Large mode volume integrated Brillouin lasers for scalable ultra-low linewidth and high power

Kaikai Liu ®[1], Karl D. Nelson[2], Ryan O. Behunin[3,4] & Daniel J. Blumenthal ®[1] ✉

Ultra-low linewidth, high output power, integrated single mode lasers, that operate from the visible to shortwave-IR, are critical for future compact, portable, precision applications. Achieving this performance in a CMOS compatible integration platform that can also enable scaling to lower linewidths and higher powers remains a key challenge. We report demonstration of a class of integrated laser with a 31 mHz instantaneous linewidth, 41 mW output power, and 73 dB sidemode suppression ratio, tunable over 22.5 nm. This performance is possible due to Brillouin nonlinear laser dynamics in a large mode volume, meter-scale, MHz free spectral range, low loss silicon nitride coil resonator with the potential to scale to an operating regime of mHz fundamental linewidth and Watt class lasers. Such lasers hold promise to unlock new sensitivity and fidelity for quantum sensing and computing, ultra-low-noise mmWave and RF generation, fiber sensing, and atomic, molecular, and optical physics.

Low fundamental linewidth and low-noise integrated lasers with high output power are critical for a wide range of precision scientific and commercial applications, including coherent communications[1], fiber-optic sensing[2], coherent laser ranging[3], atomic and quantum sensing[4], atomic clocks[5], and mmWave generation[6,7]. Specific examples include delivering precision high power light to narrow atomic transitions[8,9], generation of low phase noise mmWave and RF sources[6], and precision neutral atom gravitational time-shift measurements[10]. Novel laser designs that enable reduced linewidth and increased output power will unlock the potential for new precision applications and miniaturization, improving experimental reliability and portability. Yet today, integrated laser solutions that can deliver this class of performance have remained elusive. There is a need for integrated lasers that can achieve the operating regime of low fundamental linewidth and high output power and can provide scaling towards mHz fundamental linewidths and Watt class output power.

Lab-scale large mode volume lasers, such as fiber-based external cavity lasers (ECLs), deliver high performance via a large intra-cavity photon number and high cavity quality factor (Q) needed to reduce linewidth and increase output power. Yet these table-top lasers are highly susceptible to environmental disturbances and require complex cavity stabilization techniques, with linewidths largely limited to the Hz level regime. Photonic integration offers low linewidth solutions including ECLs[11–14], self-injection locking lasers (SIL)[15–17], and stimulated Brillouin scattering (SBS) lasers[18–23]. State of the art integrated ECLs are capable of output powers in the range of 20 mW with the fundamental linewidth on the order of 10 Hz[11,12]. Large mode volume table-top and integrated ECLs can approach Hz-level fundamental linewidths and can employ external lab-scale or integrated[24] resonator reference cavities in a reflection mode to further reduce integral linewidths, where the reference is a cold-cavity and the resulting high frequency noise defined by the single mode laser that is stabilized to one reference cavity resonance. With these cold-cavity stabilized lasers, the large reference cavity mode volume is used to reduce the thermorefractive noise (TRN) floor and associated close to carrier and mid-carrier offset frequencies (e.g., 1 Hz–1 kHz). However, using a cold reference cavity in the transmission mode to reduce fundamental linewidth, e.g., offset frequencies beyond

[1]Department of Electrical and Computer Engineering, University of California Santa Barbara, Santa Barbara, CA, USA. [2]Honeywell Aerospace Technologies, Plymouth, MN, USA. [3]Department of Applied Physics and Materials Science, Northern Arizona University, Flagstaff, Arizona, USA. [4]Center for Materials Interfaces in Research and Applications (¡MIRA!), Northern Arizona University, Flagstaff, AZ, USA. ✉e-mail: danb@ucsb.edu

100 kHz, is difficult since the low linewidth cavity technical noise translates to high frequency noise.

Lasing based on nonlinear optical feedback, such as integrated SBS and SIL lasers, are important approaches to reducing fundamental linewidth, capable of producing sub-Hz linewidths[18–21]. To date, it has been difficult to achieve both linewidth reduction and output power increase due to laser physics including mode saturation, the onset of higher order modes, and limited cavity mode volume and the resulting modified Schawlow-Townes linewidth (STLW) and TRN limits. In terms of the nonlinear feedback physics, there is a fundamental difference between SIL and SBS. The SIL laser augments an already single lasing mode using, e.g., a distributed feedback (DFB) or distributed Bragg reflector (DBR) based design, or the collapsing of a multimode Fabry Perot (FP) laser into a single mode using a high reflectivity feedback cavity. Notably, SIL utilizes highly nonlinear feedback to provide noise suppression in an already lasing semiconductor resonator. SBS lasers utilize a fundamentally different nonlinear mechanism, where an external laser is used to generate phonons in a resonator and reduce optical frequency noise through highly nonlinear feedback between input (pump) and output photons (S1) and the intermediary phonons. Yet, achieving both high output power and narrow linewidths in integrated SBS lasers has been challenging. To date, these lasers employ small cavity mode volumes with an FSR set by the phase matching condition between cavity resonance and the Brillouin frequency shift. The requirement for single mode operation has driven SBS laser cavity designs to support one or multiple FSR per Stokes frequency shift from the pump where the Brillouin gain bandwidth overlaps with only a single cavity resonance[18–21]. This design inherently leads to cascaded emission, which limits the output power and linewidth narrowing[25]. Designs that inhibit the onset of second order Stokes lasing have been employed to decrease the linewidth and increase the single mode output power[26–28]. However, in these designs the first-order Stokes (S1) laser emission grows modestly− scaling with the square-root of the pump power−limiting the output power and the linewidth[25,28]. Additionally, the small mode volume further limits the linewidth through the TRN floor which scales inversely with the optical mode volume[29,30] and ultimately the modified STLW. While increasing the resonator quality factor (Q) of these integrated SBS cavities has enabled linewidth reduction[18,19,21], the resulting low-threshold coupled with cascaded emission limit the linewidth and the output power. Therefore, new solutions are needed to overcome these limits and enable scalable reduction in linewidth and increase in optical output power.

In this work we report a novel integrated SBS laser that can simultaneously achieve high output power and low fundamental linewidth into a spectrally pure single frequency mode. Our laser utilizes an active meter-scale coil waveguide resonator to drive down the linewidth and drive up optical power through an increase in intracavity photons, reduction in the TRN floor, and an increase in the S1 optical power saturation level. The laser outputs a single mode 31 mHz fundamental linewidth with 41 mW optical power and strong single mode operation with 73 dB sidemode suppression ratio (SMSR) using an active 160 million intrinsic Q meter-scale coil-resonator cavity (4-meters). We further demonstrate that this laser can be Vernier tuned across a 22.5 nm range. This work differs fundamentally from prior works, in terms of operation and underlying physics, that employ meter-scale cold-cavity references for integral linewidth reduction[24]. In the present work, the resonator is the nonlinear laser cavity itself, where Brillouin physics[20,25] inside the cavity provides a down selection of both the pump laser photons and the vacuum driven spontaneous emission modes. The Brillouin nonlinear process is so selective that vacuum driven spontaneous modes within the primary resonance and all other cavity resonances that overlap the Brillouin gain are excluded from utilizing pump photons for scattering except for the very select modes. This property is experimentally measured and demonstrated. Remarkably, although the $Si_3N_4$ waveguide meter-scale coil resonator

is a multimode cavity with 48.1 MHz FSR that supports 5 longitudinal cavity modes across the Brillouin gain bandwidth, the strong nonlinear Brillouin photon-phonon feedback process demonstrates such high selectivity that the output mode measures greater than 70 dB SMSR. This process is defined by the strength and frequency selectivity of the dominant Brillouin grating that scatters the output Stokes light. In other words, the highly nonlinear physics of the dominant Brillouin grating (phonon) formation in the coil resonator itself "steals" all of the pump photons to drive the single lasing mode. This "winner take all," out-competes the vacuum driven spontaneous modes within the primary resonance and the other resonances that overlap the Brillouin gain bandwidth. This lasing principle is fundamentally different than approaches that use a meter-scale coil resonator as an external stabilization cavity to reduce the close to carrier noise for a separate laser operating above threshold[24] that does not drive down fundamental linewidth and does not provide an increase in optical output power. As a result, the Brillouin dynamics in our large mode volume, ultra-high Q cavity, can scale both linewidth and power. We discuss that a path forward with increasing cavity length can be extended to drive the fundamental linewidth down to 1 mHz or lower and output power to above 1 Watt. Owing to its compact footprint and efficiency, this approach to narrow linewidth and high-power integrated lasers can enable portable new forms of precision applications spanning from visible to NIR wavelengths.

## Results
### Principle of operation

The principle of operation and the large mode volume SBS laser configuration are illustrated in Fig. 1. A 4-meter long laser cavity is implemented as an ultra-low loss silicon nitride ($Si_3N_4$) waveguide (Fig. 1a) coiled into a bus-coupled resonator on a chip smaller than a square centimeter (Fig. 1b). A critical point is that the Brillouin laser cavity can be implemented in a complex waveguide geometry such as the coil resonator to the fact that the photon-phonon scattering interaction occurs over the length of the resonator only using low loss photon guiding and without phonon guiding[20,25]. Below threshold the Brillouin process is dominated by filtered spontaneous emission, where pump photons are scattered randomly in a manner that preferentially populates the Stokes modes that lie within the gain bandwidth (see Fig. 1). As the pump power increases, the beat note of pump light with the spontaneous emission form stress-optic gratings that enhance the backscattering of frequency shifted Stokes photons at the cavity resonances. The Stokes mode with the largest emitted power, forms the deepest stress-optic grating, further enhancing the emitted power. As a result of this form of feedback, the Stokes mode with the largest gain (i.e., emitted power) reaches threshold first, where the SBS gain is precisely balanced by the loss, and the intracavity pump power is clamped. As a consequence, the emission of the subthreshold Stokes modes is fixed and only the power of the lasing mode increases as the supplied power increases.

The broad Brillouin gain spectrum of ~250 MHz at 1550 nm spans the 4 coil cavity FSRs (Fig. 1c). The physics of this Brillouin laser does not require acoustic guiding, allowing the resonator design to focus on the waveguide design for low propagation loss. Below threshold, the pump is spontaneously back-scattered by a thermal population of phase-matched phonons into the four optical resonances that lie within the Brillouin gain bandwidth (illustrated by the orange, red, green, and purple acoustic gratings in Fig. 1d). The scattered power in these four optical modes continues to grow with increasing pump until threshold is met for the mode with the largest optical gain (S1). Above threshold the intracavity pump power is clamped (Fig. 1e), and only the lasing mode S1 increases with supplied power (purple in middle of Fig. 1f). Consequently, the other sub-threshold modes within the SBS gain bandwidth that are driven by the pump are clamped, remaining fixed in power (orange, red, green in center of Fig. 1f). The result is

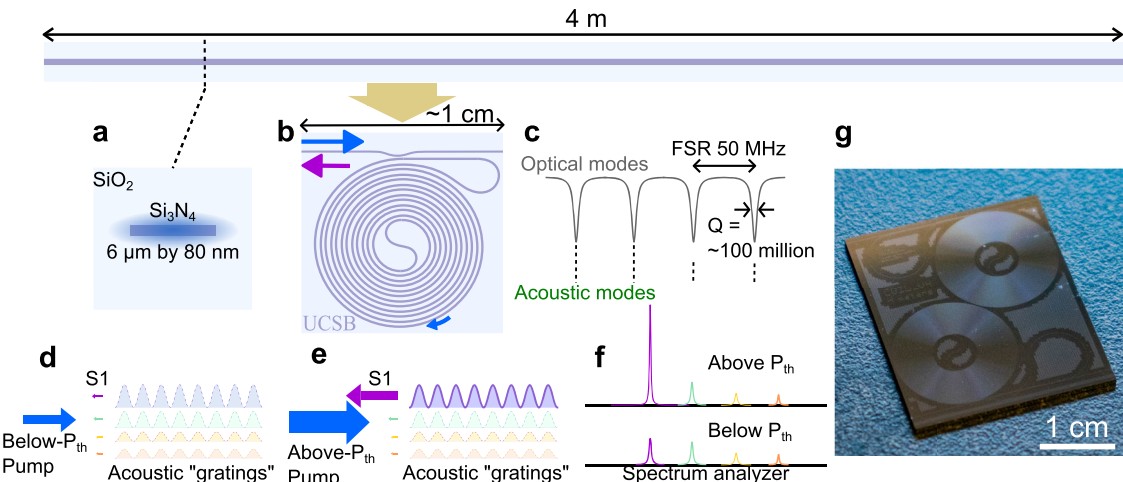

**Fig. 1 | Large mode volume SBS laser design and operation principles. a** 6 $\mu$m by 80 nm Si$_3$N$_4$ waveguide design supports the fundamental TE mode with a minimum bending radius of -1 mm and propagation loss of -0.2 dB/m in C and L band. **b** Large mode volume SBS laser implemented in a 4-meter-coil resonator. **c** Coil resonator with 50 MHz FSR overlaps with the broad spontaneous acoustic mode continuum of the Brillouin gain bandwidth (green). **d**, **e** Principle of single mode operation in a multi-mode laser cavity for below (**d**) and above (**e**) Brillouin lasing threshold. Below threshold the pump scatters uniformly in gratings formed at each of the cavity resonance that overlap with the Brillouin gain. Above threshold, the grating at the resonator mode that overlaps with the Brillouin gain peak rapidly builds up leading to single mode lasing in a large mode volume, normally multi-mode laser resonator. **f** Lasing modes below and above threshold. **g** A photograph of the fabricated 4-meter-coil resonator device. SiO$_2$, silicon dioxide. Si$_3$N$_4$, silicon nitride. SBS, Stimulated Brillouin scattering. FSR, free spectral range. TE, Transverse electric. Q, quality factor. S1, first-order Stokes. P$_{th}$, optical threshold power.

illustrated in Fig. 1d–f with single mode operation in a large mode volume and traditionally multimode resonator. The nonlinear feedback process of stimulated Brillouin scattering is so dominant, the lasing mode also takes away pump photons from the build-up of spontaneous modes (acoustic gratings or phonons) within the lasing resonance itself, which leads to very strong linewidth narrowing that leads to orders of magnitude smaller laser (S1) linewidth than the cavity resonance[25].

## Coil SBS laser design

The 4-meter-coil resonator design sets the foundation of the SBS laser. For a large mode volume SBS laser, due to the cascaded Stokes emission nature of SBS lasers in general, it is desirable to operate the pump power just below the second-order Stokes (S2) threshold. For the design demonstrated here, the S2 threshold is expected to be 4 times the S1 threshold. Driving the pump up to just below the onset of S2 lasing is the point where the S1 fundamental linewidth reaches a minimum[19,25,28]. The cavity external coupling Q ($Q_{ex}$), loaded Q ($Q_L$) and the cavity Brillouin gain ($\mu$) determine the S1 threshold, output power, and the minimum fundamental linewidth[19,25,28]. The S1 threshold ($P_{th}$) and the S1 output power at its power clamping point are linearly proportional to the cavity length while the corresponding minimal fundamental linewidth ($\Delta\nu_{min}$) reduces linearly with the cavity length ($L$). In this mode of operation, it is desirable to increase the optical mode volume by increasing the cavity length, which in turn increases the S1 output power and reduces the fundamental linewidth. The waveguide loss and coil resonator Q are important parameters, since SBS threshold is reduced as the cavity Q increases.

For the coil resonator we employ a 6 $\mu$m wide by 80 nm thick Si$_3$N$_4$ core with silica cladding waveguide design. This waveguide supports the fundamental transverse-electric (TE$_0$) mode with moderate waveguide confinement, propagation loss around 0.2 dB/m in C and L band, and a critical bending radius less than 1 mm (see the waveguide design in Supplementary Fig. S1). By leveraging this tight bending radius, a 4-meter-long waveguide resonator can be realized in a penny-size footprint (see the photograph in Fig. 1g). The resulting coil resonator has a 48.1 MHz FSR and 160 million intrinsic Q in the C and L band (see the waveguide loss and resonator Q measurements in

Supplementary Fig S3). Within the coil, neighboring waveguides are spaced by 30 $\mu$m and the minimum radius of curvature is -1.8 mm to avoid bending losses. The spectral response in Supplementary Fig. S3a shows the TE0 resonance only. Although the high-aspect-ratio waveguide is a multimode waveguide, the coil resonator center S bend acts as a modal filter that is lossy for higher-order modes and low loss for the TE0 and TE1 modes[24]. Additionally, the bus-resonator coupling is designed to couple only the TE0 mode in C and L bands to achieve a quasi-single-mode resonator[21,31]. A comprehensive review of the design considerations of coil resonators can be found in previous work[24,31]. The device fabrication details can be found in Methods as well as in previous work[21,32]. The TE$_0$ mode waveguide losses are measured by spectral scanning the coil resonator with the laser detuning calibrated by a fiber Mach-Zehnder interferometer (MZI). The lowest loss is measured to be -0.2 dB/m from 1550 nm to 1630 nm in L band while below 1550 nm the loss increases due to the N-H absorption at 1520 nm[31,33]. For better fiber-to-chip coupling, the bus waveguide is tapered from 6 $\mu$m to 11 $\mu$m. Ultra-high numerical aperture (UHNA) fibers with a mode field diameter of 4.1 $\mu$m are used to fiber-pigtail the tapered waveguide facets. The mode-overlapping simulation (shown in Supplementary Fig. S2) shows a theoretical coupling loss between the UHNA fiber and the 11 $\mu$m tapered waveguide of 1.9 dB per facet and 3.8 dB total. The fiber pigtailed connection results in a total coupling loss of 5.8 dB (see photo of packaged coil resonator in Supplementary Fig. S1).

## Coil SBS laser characterization

The SBS laser characterization is featured with the achievement of high output power and narrow fundamental linewidths. The results shown in Fig. 2 were obtained using a widely tunable external cavity laser as the SBS pump which is amplified by an erbium doped fiber amplifier (EDFA) at the resonator, then Pound-Drever-Hall (PDH) locked to the coil resonator near 1570 nm. The S1 SBS emission propagating in the opposite direction to the pump is collected using a fiber circulator located at the chip input. The S1 emission is photo-mixed with the pump on a fast photodetector to generate a beatnote to resolve the S1 power spectrum in the radio-frequency domain on an electric spectral analyzer (ESA). Just below threshold, in the spontaneous Brillouin

emission regime, multiple Stokes tones are observed, which resembles the simulated Brillouin gain profile (see the "Method" of the Brillouin gain simulation in refs. 20,22). Above threshold, we observe rapid emergence of the main Stokes lasing mode (S1) with a 73 dB SMSR (Fig. 2). Also observed is the further Brillouin linewidth reduction that occurs at the primary cavity mode. A coupled-mode model (described in Supplementary Note 2) shows that this single-mode lasing results from clamping of the pump once threshold is met for the Stokes mode with the largest gain, which inhibits competing optical modes from reaching threshold. After confirming the property of single mode lasing, we increase the pump power to increase the S1 output power, which is measured on an optical spectral analyzer (OSA), (see Supplementary Fig. S4). The on-chip S1 output power is calibrated and plotted versus the pump power (inset in Fig. 3 and Supplementary Fig. S4). We measure an S1 threshold of 72 mW and generation of 41 mW on-chip S1 power at a 242 mW pump power, which corresponds

to an overall 16.5% efficiency (Supplementary Fig. S4). The slope efficiency is measured to be 36%. With the 2.9 dB facet fiber coupling loss, there is -13 dBm measured optical output power in the fiber.

The laser frequency noise and fundamental linewidth are measured using an unbalanced fiber optic MZI with a 1.037 MHz FSR as an optical frequency discriminator (OFD) and the delayed self-homodyne laser frequency noise measurement method as described in Methods[20,34]. The fundamental linewidth is calculated from the white-frequency-noise floor of the frequency noise spectrum[20,28], $\Delta\nu_F = \pi S_w$. To enable measurement of such low frequency noise and fundamental linewidth we characterize the OFD frequency noise floor that starts to emerge above 10 MHz frequency offset and is primarily from the balanced photodetector used in the OFD (light blue dash in Fig. 3). The red frequency noise curve in Fig. 3 measured 10 mHz²/Hz frequency noise at high frequency offsets and is used to calculate the fundamental linewidth of 31 mHz.

It is worth noting that the coil SBS laser frequency noise reaches the coil resonator's TRN limit at frequency offsets from 10 kHz to 100 kHz. Below 10 kHz the S1 frequency noise is dominated by environmental noise such as vibrations and acoustic noise, as well as photothermal noise due to the intracavity power fluctuation induced by the pump laser intensity noise[35,36]. The low to mid frequency pump noise is reduced towards the TRN floor when PDH locking the pump to the coil resonator (green trace in Fig. 3). The Brillouin process further reduces the frequency noise by several orders of magnitude above several hundred kHz where the pump-transferred frequency noise gets suppressed with the increasing pump power[31]. Above 10 MHz the measured frequency noise is dominated by photodetector noise (dashed light blue trace in Fig. 3). Although we can measure a fundamental linewidth of 31 mHz, the estimated fundamental linewidth in this 4-meter-coil SBS laser operating at the S1 clamping point is lower than the measured value. In previous work[20,37], we measured an SBS laser fundamental linewidth of 0.7 Hz at the S1 clamping point in a 11.83-mm-radius ring resonator with a cavity length of 74.3 mm. Here, the 4-meter-coil cavity length is 54 times the ring resonator cavity length, resulting in the fundamental linewidth expected to decrease by 54 times to 13 mHz. The ability to measure low magnitude frequency noise and fundamental linewidth at 0.1 MHz and above is limited by the frequency noise measurement system. The ability to reliably make these measurements are a subject of future work. Above 10 MHz the OFD frequency noise measurement is limited by the photodetector noise (light blue dashed curve in Fig. 3). Given the existence of the excess noise from the OFD frequency noise measurement (e.g., servo bandwidth and quality factor, MZI FSR, and PD noise floor) and

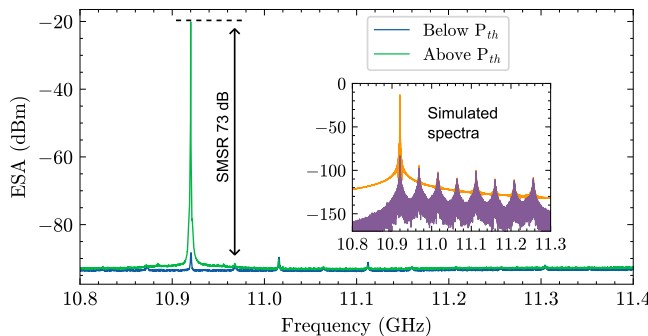

**Fig. 2 | Coil SBS laser single mode lasing.** Pump-S1 beatnote spectrum of the SBS laser at 1570 nm on an ESA in spontaneous emission below threshold (blue trace) and stimulated emission above threshold (green trace) demonstrates 73 dB SMSR. Inset shows a simulated spectra of the Brillouin grating modes at cavity resonances that overlap Brillouin gain spectrum below-threshold (blue trace) and above-threshold (green trace). See Supplementary Information Supplementary Note 2: SBS laser coupled mode model and single-mode lasing for further details on the underlying theory and simulation details. Measurements show pump energy is driven to the dominant lasing mode above threshold with a reduction in gratings that can contribute to unwanted side mode power. Also shown is linewidth reduction in the dominant mode above lasing due to successful competition for pump photons within the dominant mode resonator linewidth. ESA, electrical spectrum analyzer. $P_{th}$, SBS threshold. SBS, Stimulated Brillouin scattering. SMSR, side mode suppression ratio, $G_B$, Brillouin gain spectrum.

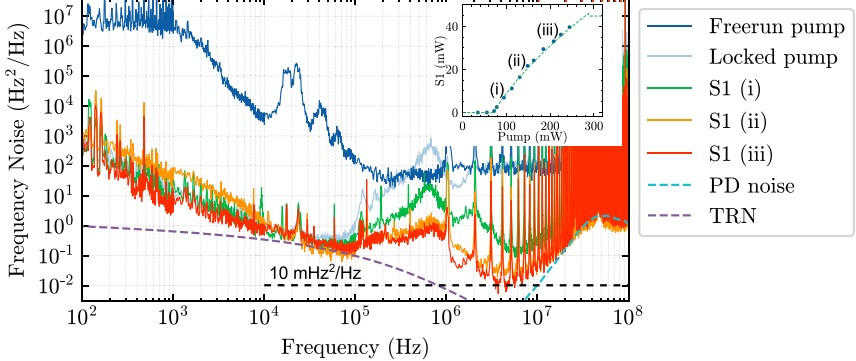

**Fig. 3 | SBS laser output power, frequency noise, and fundamental linewidth.** OFD frequency noise measurements for the SBS laser at different pump powers show the minimum fundamental linewidth of 31 mHz, corresponding to a frequency noise of 10 mHz²/Hz. At frequency offset above 10 MHz, the OFD frequency noise measurement is limited by a noise floor from the photodetector. Inset shows the SBS laser on-chip power versus on-chip pump power with a threshold of 72 mW and an output power of 41 mW. Purple dashed curve is the simulated TRN of the coil resonator. At frequency offsets above 10 MHz, the OFD frequency noise measurement is limited by the photodetector noise, as indicated by the light green dashed curve. OFD, optical frequency discriminator. PD, photodetector. TRN, thermo-refractive noise.

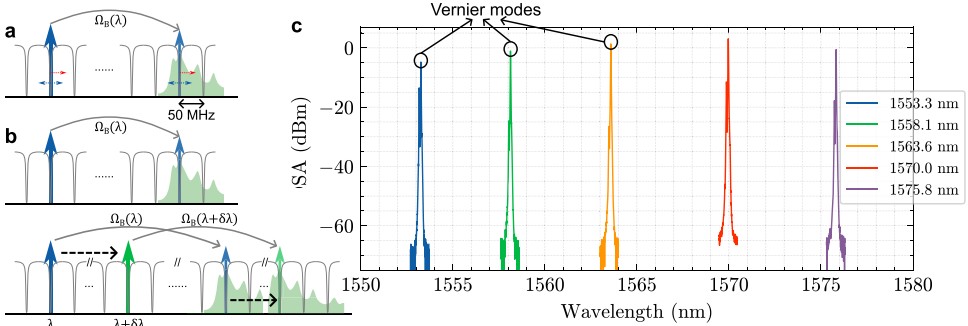

**Fig. 4 | Wide-wavelength-range Vernier tuning in a coil SBS laser in C and L band. a** The SBS phase matching condition is satisfied for a Brillouin shift of ~10.6 GHz. The Brillouin gain bandwidth is about 250 MHz wide due to the non-confined phonons (bulk-like). Thermal tuning of the cavity to less than an FSR provides fine tuning of the SBS output (first order Stokes S1) as demonstrated in prior work. **b** The Brillouin output can be medium tuned by adjusting the pump over a small number of FSR which will track the Brillouin gain (not demonstrated in this work). For wide tuning, the Vernier effect is demonstrated (lower illustration), where the Brillouin shift $\Omega_B$ is linearly proportional to the optical frequency and changes by more than an FSR when the pump wavelength changes by ~6 nm around 1570 nm. **c** S1 emission recorded on the OSA reaches a local minimal threshold every ~6 nm in C and L band. OSA, optical spectrum analyzer. $\Omega_B$, Brillouin shift frequency. FSR, Free spectral range. SBS, Stimulated Brillouin Scattering. OSA, Optical spectrum analyzer.

residual pump-transferred noise, the claim of the 31 mHz fundamental linewidth represents an upper bound of the fundamental linewidth. The OFD frequency noise measurement is optimized to minimize the measurement noise as described in detail in the previous work[28] and in Methods.

We additionally demonstrate that this approach can be used to realize a broadly tunable Vernier tunable SBS laser. The SBS laser can be fine-tuned around each lasing resonance (less than an FSR) by adjusting the coil resonator temperature as has been previously shown[38]. By tuning the pump to a neighboring resonance (one FSR away), the laser output can be tuned to a neighboring 48 MHz resonance as long as the wavelength dependent Brillouin frequency shift remains small compared to the FSR. This condition can hold for a pump tuning over a small number of FSRs. When the pump laser is tuned by more than a small number of resonances, there is a pump wavelength dependent Brillouin gain frequency shift that occurs, causing the gain to overlap with a resonance that is a different integer number of FSRs away from the pump, creating a Vernier condition. This occurs since the since the dependence of the Brillouin frequency shift on the pump mode wavelength is $\Omega_B \propto 1/\lambda$, and since the coil resonator FSR is smaller than the Brillouin gain bandwidth (in this case ~50 MHz FSR compared to 250 MHz), the phase matching condition takes on a Vernier relation, in this case every ~6 nm. This is illustrated in Fig. 4a where the SBS phase matching condition $\Omega_B(\lambda) = N\nu_{FSR}$ is satisfied at one wavelength point ($\lambda$) and for a second wavelength point ($\lambda + \delta\lambda$) the SBS-phase-matched wavelength occurs at $\Omega_B(\lambda + \delta\lambda) = (N - 1)\nu_{FSR}$ is $\delta\lambda = \lambda\,\nu_{FSR}/\Omega_B$. For this design, this Vernier relation is ~6 nm around 1570 nm operation. This behavior is enabled by the small FSR (due to the long coil length) compared to the 250 MHz gain bandwidth, a condition which is not possible for waveguide ring resonators with FSRs of a few GHz. The dependence of FSR on wavelength (coil resonator cavity dispersion) is negligible compared to the acoustic frequency dependence on wavelength. The OSA traces of the S1 emission at the on-chip pump power of ~140 mW at different wavelengths from 1550 nm to 1580 nm (Fig. 4b) are recorded when S1 reaches a local minimal threshold and local maximal output power. From 1570 nm to shorter wavelengths, at the phase-matched wavelengths, the SBS threshold increases due to increased waveguide losses and decreased cavity Qs (see Supplementary Figs. S3 and S5). It has been reported in literature that in between the optimally phase-matched wavelengths, where the lasing mode is not on the peak of the Brillouin gain, the SBS laser threshold increases and fundamental linewidth increases due to the increase in the linewidth enhancement factor[21,39]. Although above 1570 nm the SBS threshold is estimated to

decrease, the C-band EDFA does not provide optical gain and amplification above 1580 nm to pump the SBS laser. With the technique of two-point coupling that provides resonator coupling in large wavelength range, it is possible in the future work to achieve an SBS laser in one single coil waveguide resonator at multiple wavelength bands such as C, L, O bands, and even visible light[23,31].

## Discussion

We have demonstrated a photonic integrated coil-Brillouin laser that can simultaneously achieve ultra-narrow linewidth single mode operation and high emitted power. The design employs Brillouin lasing in a large mode volume, meter-scale coil resonator cavity to achieve 31 mHz fundamental linewidth, 41 mW output power, a 73 dB SMSR, and Vernier tuning across a 22.5 nm range. This performance corresponds to greater than 5 orders magnitude frequency noise reduction from the free running pump laser over a wide frequency noise range. These results are made possible by achieving single mode Brillouin lasing in a 4-meter-long coil resonator cavity with 160 million intrinsic Q.

Here we report an active Brillouin parametric gain medium in the coil resonator, that differs from prior coil-based work[32], in that the nonlinearity converts pump photons with high efficiency to output Stokes signal with the reported phase noise and linewidth properties, without the onset of multimode lasing. The result is the lowering of mid- to high-frequency noise through the nonlinear Brillouin process that converts pump photons to the first order Stokes with such high efficiency that the other resonator modes, and most spontaneous modes that overlap the Brillouin gain spectrum, are not able to utilize the pump photons as the power is increased. This process in an increased mode volume coil resonator with an ultra-high Q coil achieves the record performance through the continued increased photon number and output power before the onset of second order Stokes, hence continuing to drive down the ST fundamental linewidth. At the same time the highly efficient Brillouin nonlinear process prevents multimode operation even though multiple cavity resonances overlap with the Brillouin gain bandwidth. The result is to significantly lower the high frequency noise through Brillouin and the mid-frequency noise through reduced TRN floor due to the large cavity mode volume[35,36]. In stabilized PDH lock laser configurations[32] the coil resonator is used as a cold reference cavity, where a small amount of laser power is tapped and input to the coil together with a PDH lock loop of finite bandwidth (e.g., 100 kHz or lower) to stabilize the laser to the cold cavity. The PDH technique using a cold cavity coil brings down the low- to mid-frequency noise with a lowered TRN and photothermal noise due to low optical power. The bandwidth of the noise reduction

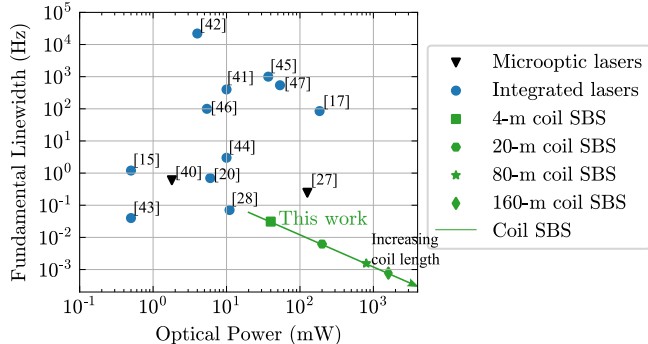

**Fig. 5 | Scaling the integrated coil Brillouin laser to 1 mHz fundamental linewidth and greater than 1 Watt output power.** The green line indicates what is possible with the coil SBS lasers, and by increasing coil lengths the SBS laser output power at the S1 clamping point increases linearly with the coil lengths while the fundamental linewidth decreases linearly with the coil length, assuming that the coil resonator Qs stay relatively the same. To reach below 1 mHz fundamental using coil SBS lasers, the coil length needs to be increased by 30 times from 4 meters to 120 meters, and the output power can be increased to 1200 mW.

is limited to the bandwidth and phase response of the net optoelectronic feedback loop as well as the presence of a servo-bump at the loop bandwidth in the noise spectrum. Hence, the PDH locked cold cavity approach does not lower the high frequency noise. The two techniques nonlinear high-frequency noise reduction using the active coil-resonator and low- to mid-frequency noise reduction using a PDH lock to a passive coil-resonator address very different parts of the frequency noise spectrum. The two approaches can be combined where a coil-Brillouin laser is PDH locked to a cold coil reference cavity to mitigate the noise across the frequency spectrum[36], which for the coil Brillouin laser is a subject of future work. This current work is also fundamentally different from a prior reported sub-Hz integrated Brillouin laser[20], where the linewidth in that work was limited by several factors. First, the relatively small mode volume of the laser cavity set a limit on the ST fundamental linewidth due to limited first order Stokes power and number of photons. Second, the onset of second order Stokes lasing clamps the number of photons in the first order Stokes and injects phase noise back into the first order, driving up the fundamental linewidth in the first order[25]. Third, the single longitudinal mode cavity of the smaller cavity[20] sampled the Brillouin gain with a single FSR resonance, limiting the cavity length, mode volume, first order Stokes output power, and fundamental linewidth. The current coil Brillouin laser results eliminate these bottlenecks by enabling an increasing length (hence mode volume) cavity without the onset of multimode lasing and enabling the fundamental linewidth to be continued to drive down lower without noise being fed back from the onset of second order Stokes, and without the onset of multimode lasing. This result allows the linewidth to scale lower and output power to scale higher due to the Brillouin nonlinear gain producing a single mode laser in a multimode cavity in an integrated waveguide laser.

Remarkably, the Brillouin lasing physics allows the use of a large mode nonlinear photon-phonon resonator with multi-FSR per gain bandwidth to achieve high SMSR single mode lasing. Using a coupled mode model, and verified experimentally, we show that single mode lasing is promoted for the mode with the largest Brillouin gain. The lasing mode that reaches threshold first prevents vacuum driven spontaneous modes within the primary resonance and from other cavity resonances from reaching stimulated emission. To support these conclusions, we measure the frequency noise from below threshold, through threshold, up to saturation of the S1 mode, demonstrating linewidth reduction and increased coherence. The physics here differ markedly from approaches that employ such resonators as cold-cavity references where the lasing response is

defined by the external semiconductor laser that interacts with only one cavity resonance and a PDH loop. In this work, the resonator is the nonlinear laser cavity itself, where Brillouin physics forms a process that is so selective, the resulting down selection of pump laser photons as well as vacuum driven spontaneous emission modes in the Brillouin nonlinear dynamics, are capable of forming the 30 mHz linewidth, 40 mW lasing output. In the case of lasing mode selection and robustness, the laser will select the overlap between the highest Brillouin gain frequency and its nearest overlapping resonator mode. Due to the large Brillouin gain and wide gain bandwidth provided by non-guided phonon modes[16], the mode selection and SMSR are robust to exact positioning of the cavity resonance relative to the gain, as is demonstrated by Vernier tuning as described further in Section. The Brillouin selection process is so strong that the main lasing mode inhibits spontaneous emission within a single resonance. This fundamental property of the Brillouin nonlinearity will allow this design to continue to scale to longer cavities with smaller FSR. With temperature changes and fluctuations, the Brillouin gain center frequency is insensitive to temperature[16] and a change in the FSR will result in wavelength tuning or Vernier tuning. Such changes may be desirable, as is the case with a tunable Brillouin laser, or would require temperature stabilization of the chip or athermal resonator designs.

As a note on Brillouin lasing in comparison to fiber implementations. The broadened gain bandwidth due to our non-guided phonons is not primarily responsible for the highly selective nature. It is the large gain and feedback of the Brillouin process coupled with the ability to make very stable long Brillouin gain and resonator cavities on chip, and large single pass cavities with high Q. This leads to very high effective length via the coil length and Q over which the Brillouin interaction can take place which is difficult to do stably in fiber. Fiber Brillouin lasers employ linewidth narrowing based on the same physics, although the phonon modes are quite different in the planar and cylindrical structures, yet as far as we are aware this description of the process has not been described before. There is also the possibility that the low loss, high Q, and large effective length of the integrated coil resonators leverages the long photon coherence to phase lock the bulk phonon modes all along the resonator, creating a super grating over the whole structure that is highly frequency selective. These topics are the subject of current and future work.

In Fig. 5 we provide a comparison of this laser performance to other integrated and non-integrated micro-optic lasers including SIL and Brillouin lasers[15,17,20,27,28,40–47]. Also shown in Fig. 5 is that by scaling the coil to 20 meters and beyond, in combination with the Brillouin lasers, this design can decrease the fundamental linewidth down to 1 millihertz fundamental linewidth and output power exceeding 1 Watt. Coil lengths of 4 meters to order 20 meters have been demonstrated today, and longer coils out to 200 meters could be possible using multi-layer silicon nitride waveguides[8,9]. Notably, although the fiber based cavity can have similar or larger optical mode volume than the integrated waveguide coil resonators, the single-frequency fiber Brillouin lasers based on the 2-meter-long and 11-meter-long fiber cavities[10,48] are reported to have measured fundamental linewidths of 7 Hz and 150 Hz, respectively, and optical output powers of 9 mW and 0.19 mW, respectively. Such fiber-based lasers are subject to environment driven fluctuations and noise sources that require complex isolation and thermal and piezoelectric feedback control. We also have demonstrated that through the Vernier effect, we can tune the laser output wavelength in discrete steps over a range of 22.5 nm. This grid of accessible wavelengths is due to the meter-scale coil FSR of 48.1 MHz and the SBS phase matching condition being satisfied every ~5.5 nm. In order to reduce the Vernier period in wavelength, the most straightforward approach is to reduce the FSR by increasing the coil resonator length, which comes at an expense of increasing the laser threshold as it increases with the coil resonator length. This is the subject of future work. With the technique of two-point coupling that

provides resonator coupling in large wavelength range, it is possible in the future work to achieve an SBS laser in one single coil waveguide resonator at multiple wavelength bands such as C, L, O bands, and even visible light[23,31]. These large mode volume SBS coil lasers are fabricated using a 200-mm CMOS foundry compatible process, demonstrating the potential for low cost, wafer-scale fabrication of compact precision high-performance lasers that can be integrated with other components for systems-on-chip solutions for fiber communications wavelength ultra-low phase applications such as mmWave, RF, and fiber sensing. Additionally, the ability to fabricate these coil resonators and SBS lasers in the visible[22,49] shows a path towards ultra-low fundamental linewidth, high power lasers for portable and increased reliability atomic and quantum experiments and metrology applications.

## Methods

### Fabrication process

A 15-μm-thick thermal oxide lower cladding layer is grown on a 200-mm diameter silicon substrate wafer. A 75-nm-thick $Si_3N_4$ film is deposited on the thermal oxide using low-pressure chemical vapor deposition (LPCVD), followed by a standard deep ultraviolet (DUV) photoresist spinning, DUV stepper patterning and dry etching in an inductively coupled plasma etcher using $CHF_3/CF_4/O_2$ chemistry. After etching, Following the etch, a standard Radio Corporation of America (RCA) cleaning process is applied. An additional $Si_3N_4$ thin layer is deposited followed by a 30-minute and 1100 °C anneal in an oxygen atmosphere. Lastly, a 6-μm-thick silicon dioxide upper cladding layer is deposited using plasma-enhanced chemical vapor deposition (PECVD) with tetraethoxysilane (TEOS) as a precursor, followed by a final two-step anneal at 1050 °C for 7 h and 1150 °C for 2 h.

### Laser frequency noise and fundamental linewidth measurements

A 200-m fiber MZI with a FSR of 1.03 MHz is used as an optical frequency discriminator (OFD)[20,28,34] with the two fiber outputs detected with a Thorlabs balanced photodetector (PDB 450 C, transimpedance gain $10^4$ V/A) and the balance detection signal sampled by a Keysight DOXS1024 digital oscilloscope at different sampling rates ($10^4$ Sa/s, $10^5$ Sa/s, $10^6$ Sa/s, $10^7$ Sa/s, $10^8$ Sa/s) with 10,000 sampling points at each sample rate. The power spectrum of the balanced detection signal $S_{BPD}(f)$ is converted into a laser noise spectrum $S_\nu(f)$ by[20,28] Eq. 1 below

$$S_\nu(f) = S_{BPD}(f)\left(\frac{f}{\sin(\pi f \tau_D)V_{pp}}\right)^2, \quad (1)$$

where $1/\tau_D$ is the fiber MZI FSR and $V_{pp}$ is the peak-to-peak voltage of the fiber MZI response over a full FSR detuning frequency. The balanced detection needs to be as close to perfect balancing as possible to minimize potential laser intensity noise to frequency noise conversion, and the balance-detected power is maximized for a reduced laser noise measurement noise floor, as described in more details in ref. 28.

## Data availability

The data and code that support the plots and other findings of this study are available from the corresponding authors upon request.

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

## Acknowledgements

This material is based upon work supported by DARPA GRYPHON, under Award Number HR0011-22-2-0008, and ARO AMP under Award Number W911NF-23-1-0179. The views and conclusions contained in this document are those of the authors and should not be interpreted as representing official policies of DARPA or the U.S. Government. We thank Jim Nohava, Joe Sexton, Jim Hunter, Dane Larson, Michael DeRubeis, and Jill Lindgren at Honeywell for their contributions to mask design and sample fabrication. We also thank Kwangwoong Kim at Nokia Bell Labs for fiber-pigtailing and packaging the coil resonator.

## Author contributions

K.L., R.O.B. and D.J.B. prepared the manuscript. K.L. and K.D.N. designed and fabricated the coil resonator. K.L. designed and implemented the packaging. R.O.B. simulated the Brillouin gain response and performed the theoretical modeling and analysis of Brillouin lasing. K.L. performed the SBL lasing demonstration with the OSA, ESA, and frequency noise measurements. D.J.B. supervised and led the scientific collaboration.

## Competing interests

Dr. Blumenthal's work has other work funded by Infleqtion. Dr. Blumenthal has consulted for Infleqtion and received compensation and owns stock in the company. K. Liu, K. D. Nelson, and R. O. Behunin declare no potential competing interest.
