## [Transparent Peer Review file · Nature Communications]

Large Mode Volume Integrated Brillouin Lasers for Scalable Ultra-Low Linewidth and High Power

Corresponding Author: Professor Daniel Blumenthal

Version 1:

Reviewer comments:

Reviewer #1

(Remarks to the Author)

The authors present an inspiring work on achieving the milestone 34 mHz Lorentzian linewidth of an on-chip Brillouin laser with high output power (40 mW on chip) in a thin-silicon-nitride platform that has been developed by the senior author's team and their collaborators. Although many aspects of the efforts in this work are technical after their prior series work since 2019, it indeed presents a great step towards quiet integrated lasers that can challenge fiber lasers and offers an important building block to the passive silicon nitride photonic integrated circuits. This performance is made possible by the collective efforts including Brillouin nonlinear dynamics in a large mode volume, nonlinear photon-phonon, MHz-scale-FSR, ultra-low loss silicon nitride resonator cavity. The key achievement is to simultaneously reduce the fundamental linewidth and improve the output power by one order of magnitude, compared to the state-of-the-art record made by the same group. This paper is suitable to be published by Nature Communications, after addressing the following technical concerns:

- 1) The actual photonic device should be clearly presented in Figure 1 or Figure, rather than embedded in the supplementary document.
- 2) The authors repetitively claimed that the principle of this work is fundamentally from other work using a long-coil external photonic cavity. They should also claim whether this work is fundamentally different from their work on sub-Hz integrated Brillouin laser published in Nature Photonics in 2019. This would help the readers understand the key novelty of this work, beyond the achieved technical numbers.
- 3) The 6000 nm by 80 nm silicon nitride waveguide is a multi-modal waveguide. How do the authors prevent the excitation of high-order optical transversal modes? They authors should also add experiment data to the show the optical transmission spectral response of such a 4-meter-long coil waveguide (open status) over a wide wavelength range in optical C and L bands. The authors should provide OFDR loss measurement to cross-prove the loss retrieved from the resonance linewidth measurement.
- 4) The black elliptic circle in Figure 2 should cover the actual zoomed-in range. It looks confusing.
- 5) It seems that the laser can only be tuned to several sparse wavelengths, limited by the Vernier effect. Is it correct? Can this issue be overcome? If so, please add data to show laser tuning with denser wavelength spacing.
- 6) The latest work on a high-power narrow-linewidth integrated laser should be included in Figure 5 to ensure better literature coverage.
- 7) Please correct typos in "****within an large mode volume****" (Line 23), " **modes withing**" (Line 115), "**** silica clad" (Line 179), etc.
- 8) Will the suppression of 2nd Stokes Brillouin lasing help improve the laser performance? Can you estimate it or provide any comments?

Reviewer #2

(Remarks to the Author)

In this paper, the authors demonstrated an integrated stimulated Brillouin laser with 31 mHz fundamental linewidth, 41 mW output power and 73 dB sidemode suppression ratio. This is achieved using a 4-meter coil resonator with an intrinsic Q of 160 million. The experimental results are interesting. However, before I can recommend this work for publication in Nature Communications, several concerns must be addressed.

1. The authors use a 4 meter coil resonator as the laser cavity. The length of the waveguide is comparable to the fiber-based

system. How does the performance of the demonstrated laser compare to the state-of-the-art Brillouin fiber lasers in terms of the output power, fundamental linewidth, and the thermal stability? A detailed comparison would provide valuable context for assessing the advantages and limitations of this approach.

2. If I understand correctly, achieving a high sidemode suppression ratio (SMSR) requires precise alignment of one optical resonance with the peak of the Brillouin gain profile while ensuring that the remaining three optical resonances within the Brillouin gain bandwidth fall in regions where the gain is significantly lower. This requirement would impose significant challenges as the FSR of the cavity reduces. Furthermore, as the cavity length increases, the coil cavity may become more susceptible to thermal fluctuations. Given these considerations, the authors should provide a more detailed discussions of the advantages and limitations of using a longer coil length when making claims about the scalability of this approach.

3. The authors stated that the SBS laser can be tuned every 48 MHz on the coil resonator FSR grid, however, they also mentioned that the SBS phase matching condition is only satisfied every 6 nm. Also, in Fig. 4b, SBL is only shown every 6 nm. These two claims seem contradictory. Could the authors clarify this apparent discrepancy? How does the optical spectrum would look like if the optical resonance does not perfectly locate at the highest point of the Brillouin gain profile? It would be helpful to show the optical spectrum at wavelengths beyond those presented in Fig. 4b to provide a more comprehensive understanding of the tuning behavior. Furthermore, the description of the Vernier tuning scheme is not entirely clear and would be nice if the authors can provide a more detailed explanation.

4. In Fig. 3, the authors present the frequency noise measurement of the SBLs and the free-running pump. However, the SBL is measured when the pump is locked to the cavity through the PDH loop. It would be helpful if the authors can also provide the frequency noise spectrum of the locked-pump. This additional data would provide a clearer understanding of how the SBL demonstrated here can suppress the frequency noise of the pump laser through the SBS process.

5. In the frequency noise measurements, the white noise floor of the SBL is not clearly visible, making the claim of the 31 mHz fundamental linewidth less convincing. Could the authors provide more comprehensive characterizations of the frequency noise of the SBL? Additionally, the authors employ a 200-m fiber MZI as the optical frequency discriminator, would a shorter fiber spool help to resolve the noise floor better?

Some small comments:

1. What does the purple line indicate in Fig. 5? Please clarify this in the figure legend or the main text.

2. What is the slope efficiency of the tested SBL?

3. What is the relation between the fundamental linewidth of the SBL and the pump power? How do the measured values compared to the theoretical estimation?

Reviewer #3

(Remarks to the Author)

The manuscript titled "Large Mode Volume Integrated Brillouin Lasers for Scalable Ultra-Low Linewidth and High Power" presents a high power, ultra-low linewidth laser based on Brillouin scattering. Over the last decade, Brillouin lasers have emerged as the highly coherent source for applications in Quantum technologies, Soliton combs, microwave signal generation, optical atomic clocks, and optical communications. While ultra-low linewidth Brillouin lasers have been achieved in different platforms, lack of high output power from these lasers necessitate the use of optical amplifiers, which compromises the advantage obtained from its narrow linewidth.

A high power, ultra-narrow linewidth laser with tunable operation is therefore highly desirable. The current manuscript is therefore a major step forward in achieving a source suitable for real-world applications.

The manuscript, however, lacks clear explanation and assumes that the readers will be experts in Brillouin scattering, which is not always the case. I suggest the authors to address the comments below.

Line 73: "where an external laser is used to generate phonons in a resonator...".

How does the external laser generate phonons? Typically, the pump signal first gets scattered from thermally generated phonons to create spontaneously generated Brillouin Stokes, which buildup through cavity feedback to achieve stimulated emission. In that process, beating between the pump and spontaneously generated Stokes create an acoustic wave at the beat frequency.

Line 103-107: "provides extreme down selection of both the pump laser photons and the vacuum driven spontaneous emission modes. The Brillouin nonlinear dynamics is so selective that vacuum driven spontaneous modes within the primary resonance and all other cavity resonances that overlap the Brillouin gain are excluded from utilizing pump photons for scattering except for the very select modes".

What do authors mean by “extreme down selection of ...” and “Brillouin nonlinear dynamic is so selective”. Please clarify what is extreme down selection and how Brillouin dynamics is so selective.

If the Brillouin dynamics is so selective then why it does not allow similar linewidth narrowing in other platforms. It must have the same selectivity in other platforms/devices as well. I suggest authors to clearly explain what is the reason for the selective nature of the Brillouin scattering as it is not explained well.

Line 111: The statement “This process is defined by the strength and bandwidth selectivity of the dominant Brillouin grating that scatters the output Stokes light.” may give a non-expert an impression that Brillouin gain bandwidth and selective nature of the Brillouin scattering are the same.

The authors should clearly explain the distinction between the selective nature of the Brillouin scattering and bandwidth of the Brillouin gain profile. Otherwise, it may lead to confusion because the Brillouin gain bandwidth here is 250 MHz, which is 8 times larger than the Brillouin gain bandwidth (~ 30 MHz) in an optical fiber, so why such a linewidth narrowing is not achieved in fiber-based Brillouin lasers.

The highly selective nature of SBS alone is not responsible for the narrow linewidth of the Brillouin laser, Q of the cavity resonance has a role to play here.

Authors have repeatedly used terms like “Physics of Brillouin laser”, “highly selective nature of Brillouin lasing”, “Brillouin lasing physics...” throughout the text. All of these refer to the same phenomenon yet that phenomenon has not been clearly explained. It would be good to explain it in the beginning so that it clarifies why Brillouin scattering allows selective amplification.

I will avoid use of superlatives such as extreme.

Line 131: “A critical point is that the Brillouin laser cavity can be implemented in a complex waveguide geometry such as the coil resonator owing to the physics of this type of Brillouin laser, where the photon-phonon scattering interaction occurs over the length of the resonator without using phonon guiding.”

What is the relevance of phonon guiding here?

Line 134: “The absence of phonon waveguiding leads to a broad Brillouin gain spectrum of approximately 250 MHz at 1550 nm.....”

How does the absence of phonon waveguiding leads to a broad Brillouin gain spectrum? Is phonon waveguiding present in other platforms where the linewidth is narrow? The Brillouin gain bandwidth depends on the phonon lifetime in a device. If the authors want to convey that the absence of phonon waveguiding results in a smaller phonon lifetime then please elaborate this point.

Line 174: “In this mode of operation, it is desirable to increase the optical mode volume by increasing the cavity length...”.

This statement is counterintuitive as the Brillouin lasing threshold in a microresonator is proportional to the pump mode volume. So, increasing the pump mode volume will result in increased threshold. The Brillouin threshold does not depend on the cavity Q alone but also depends directly to the mode volume. Since the Brillouin gain varies with the interaction length L according to $e^{(g_B I_P L)}$, where g_B and I_P refer to the Brillouin gain coefficient and pump intensity respectively, increasing the cavity length makes more sense.

Other comments:

Please go through the text carefully to avoid repetitions and sentence construction. Here are some examples.

Lines 68-69: “to selectively reduce the existing spontaneous in the single mode laser for the former, or drive.....”. In this sentence “existing spontaneous” should be “existing spontaneous emission”

Line 74: “nonlinear feedback between input and output photons and phonons”. What does “input and output photons and phonons” refer to in the above sentence? What are “input and output phonons” here?

Line 114: ““steals” all of the pump photons to the drive the single lasing mode....”. This sentence should be rephrased to “...pump photons to drive the single mode lasing...”

Line 115: Check Spelling of “within”

Line 206: Instead of using “dropped by a fiber circulator” it is better to use “collected using a fiber circulator”.

Line 212: “all increases in pump energy are directed...”. Check the use of “increases”

Line 277: “...Brillouin shift frequency’s dependence on the pump mode wavelength...” may be rephrased to “...dependence

of the Brillouin frequency shift on the pump mode wavelength...”

Line 278: Rephrase “to achieved”. It would be better to write the actual expression for B.

Reviewer #4

(Remarks to the Author)

K. Liu et al. demonstrate an on-chip single-mode laser with an impressive narrow linewidth of 34mHz and output power of over 40mW, tunable over 22nm. The performance is made possible due to the low-loss 4m-long Si₃N₄ integrated waveguides and represent a milestone in Brillouin lasers. The experimental results are well presented and thoroughly discussed. The authors use a precise method to measure the very narrow linewidth of the laser. In general, I recommend publication in Nature Communications because this result is important for a number of fields (not only nonlinear optics and Brillouin scattering). However, I have some comments that should be addressed beforehand.

- The same authors published already sub-Hertz Brillouin lasers (Nature Photonics 2018) and very recently a Brillouin laser with below 100mHz linewidth and 11 mW output power (Optics Letters 2024). So one criticism is: what is the fundamental breakthrough here, why is this particular laser configuration so good, is it based on the ability to fabricate so uniform, low-loss and long integrated waveguides to achieve such stable configuration?

- The authors write that the laser should have a linewidth of 13 mHz but pump-transferred noise prevents the measurement of a linewidth lower than 34mHz (one paper of the same group of authors is cited). Could the authors give some more details in this manuscript and for this laser configuration about it, too?

- About the single-mode operation: it is impressive to see that the strict phase matching condition apparently allows to enhance only one mode even if 5 modes lay within the Brillouin resonance. There is a model given in the supplementary but as far as I see, no simulations are presented.

Version 2:

Reviewer comments:

Reviewer #1

(Remarks to the Author)

The authors have well addressed the raised technical concerns, which can warrant the publication at Nature Communications. As a separate note, the integrated erbium and lithium niobate lasers should be included in the summary chart for literature coverage.

Reviewer #2

(Remarks to the Author)

The authors have answered my comments and I have no further comments. The revised manuscript can be recommended for publication.

Reviewer #3

(Remarks to the Author)

The authors have addressed my comments satisfactorily.

Reviewer #1

The authors present an inspiring work on achieving the milestone 34 mHz Lorentzian linewidth of an on-chip Brillouin laser with high output power (40 mW on chip) in a thin-silicon-nitride platform that has been developed by the senior author’s team and their collaborators. Although many aspects of the efforts in this work are technical after their prior series work since 2019, it indeed presents a great step towards quiet integrated lasers that can challenge fiber lasers and offers an important building block to the passive silicon nitride photonic integrated circuits. This performance is made possible by the collective efforts including Brillouin nonlinear dynamics in a large mode volume, nonlinear photon-phonon, MHz-scale-FSR, ultra-low loss silicon nitride resonator cavity. The key achievement is to simultaneously reduce the fundamental linewidth and improve the output power by one order of magnitude, compared to the state-of-the-art record made by the same group. This paper is suitable to be published by Nature Communications, after addressing the following technical concerns.

Author response: We would like to thank Reviewer #1 for their time and commitment in reviewing our manuscript and for the detailed feedback that makes the manuscript stronger and improves its readability. We would like to thank Reviewer #1 for recommending publication with our revised manuscript that addresses all comments per below.

1) The actual photonic device should be clearly presented in Figure 1 or Figure, rather than embedded in the supplementary document.

Author response: We have updated Fig. 1 to include a clear photograph of the device and a scale bar.

2) The authors repetitively claimed that the principle of this work is fundamentally from other work using a long-coil external photonic cavity. They should also claim whether this work is fundamentally different from their work on sub-Hz integrated Brillouin laser published in Nature Photonics in 2019. This would help the readers understand the key novelty of this work, beyond the achieved technical numbers.

Author response: Thank you for this important point, we have added a statement to the manuscript to clarify. The primary difference between this work and other coil-based work, is this work reports an active (hot) coil resonator that is pumped to create the non-linear parametric Brillouin gain medium that generates

the output Stokes signal with the reported phase noise and linewidth properties. This result is the lowering of high frequency noise through the nonlinear process that “steals” pump photons from all other resonator and most spontaneous modes. The increased mode volume of the coil resonator given by the ultra-high quality factor (Q) coil, achieves the record performance through increased photon number, increased power before the onset of second order Stokes, hence driving the ST fundamental linewidth low. At the same time the nonlinear process allows the above to happen without causing multimode operation even though multiple cavity resonances overlap with the Brillouin gain bandwidth, for the reasons explained. The result is to significantly lower the high frequency noise through Brillouin and the mid-frequency noise through reduced thermorefractive (TRN) noise floor due to the large cavity mode volume. For the other prior coil-based work, the coil resonator plays the role of a “cold” reference cavity, where a small amount of laser power is tapped and input to the cold cavity with a Pound-Drever-Hall (PDH) lock loop of finite bandwidth (e.g. 100 kHz or lower) to stabilize the laser to the cold cavity. The PDH technique using a cold cavity coil brings down the low to mid frequency noise with the lowered TRN (due to long coil) and reduce photothermal noise (PT) due to low optical power, and the bandwidth of the noise reduction is limited to the bandwidth and phase response of the net optoelectronic feedback loop as well as the presence of a servo-bump at the loop bandwidth in the noise spectrum. The PDH locked cold cavity approach does not lower the high frequency noise. These are fundamentally different techniques using passive reference cavity or active laser cavity configurations, that address very different parts of the frequency noise spectrum. The two approaches can be combined with a coil-Brillouin laser PDH locked to a cold coil reference cavity to mitigate the noise across the frequency spectrum, which is a subject of future work and outside the scope of this work. This current work is fundamentally different from the work reported in Nature Photonics 2019 “Sub-hertz fundamental linewidth photonic integrated Brillouin laser” in multiple respects. The 2019 work reported achieving sub-Hz but the linewidth was limited by several factors. First was the mode volume of the cavity limiting the ST fundamental linewidth due to limited first order Stokes power and number of photons. Second was the onset of second order Stokes lasing clamps the number of photons in the first order Stokes and actually feeds back phase noise in the first order, driving up the fundamental linewidth in the first order. Third, is that the cavity in the 2019 NP paper was a single longitudinal mode cavity, that sampled the Brillouin gain once with the FSR, limiting the cavity length, mode volume, first order Stokes output power, and fundamental linewidth. The major advance of the current work removes all of these bottlenecks by enabling an increasing length (hence mode volume) cavity without the onset of multimode lasing, enabling the fundamental linewidth to be continued to drive down lower, without the onset of second order Stokes and feedback of noise, without the onset of multimode lasing, enabling the linewidth to scale lower and output power to scale higher due to this novel process of Brillouin nonlinear gain producing a single mode laser in a multimode cavity in an integrated waveguide laser, the first time this has been realized.

Revised Text:Discussion and Conclusion

“We have demonstrated a photonic integrated coil-Brillouin laser that can simultaneously achieve ultra-narrow linewidth single mode operation and high emitted power. The design employs Brillouin lasing in a large mode volume, meter-scale coil resonator cavity to achieve 31 mHz fundamental linewidth, 41 mW output power, a 73 dB SMSR, and Vernier tuning across a 22.5 nm range. This

performance corresponds to greater than 5 orders magnitude frequency noise reduction from the free running pump laser over a wide frequency noise range. These results are made possible by achieving single mode Brillouin lasing in a 4-meter-long coil resonator cavity with 160 million intrinsic Q.

Here we report an active Brillouin parametric gain medium in the coil resonator, that differs from prior coil-based work²⁸, in that the nonlinearity converts pump photons with extremely high efficiency to output Stokes signal with the reported phase noise and linewidth properties, without the onset of multimode lasing. The result is the lowering of mid- to high-frequency noise through the nonlinear Brillouin process that converts pump photons to the first order Stokes with such high efficiency that the other resonator modes, and most spontaneous modes that overlap the Brillouin gain spectrum, are not able to utilize the pump photons as the power is increased. This process in an increased mode volume coil resonator with an ultra-high Q coil achieves the record performance through the continued increased photon number and output power before the onset of second order Stokes, hence continuing to drive down the ST fundamental linewidth. At the same time the highly efficient Brillouin nonlinear process prevents multimode operation even though multiple cavity resonances overlap with the Brillouin gain bandwidth. The result is to significantly lower the high frequency noise through Brillouin and the mid-frequency noise through reduced TRN floor due to the large cavity mode volume^{31,32}. In stabilized PDH lock laser configurations²⁸ the coil resonator is used as a “cold” reference cavity, where a small amount of laser power is tapped and input to the coil together with a Pound-Drever-Hall (PDH) lock loop of finite bandwidth (e.g. 100 kHz or lower) to stabilize the laser to the cold cavity. The PDH technique using a cold cavity coil brings down the low- to mid-frequency noise with a lowered TRN and reduce photothermal noise (PT) due to low optical power. The bandwidth of the noise reduction is limited to the bandwidth and phase response of the net optoelectronic feedback loop as well as the presence of a servo-bump at the loop bandwidth in the noise spectrum. Hence, the PDH locked cold cavity approach does not lower the high frequency noise. The two techniques nonlinear high-frequency noise reduction using the active coil-resonator and low- to mid-frequency noise reduction using a PDH lock to a passive coil-resonator address very different parts of the frequency noise spectrum. The two approaches can be combined where a coil-Brillouin laser is PDH locked to a cold coil reference cavity to mitigate the noise across the frequency spectrum³², which for the coil Brillouin laser is a subject of future work. This current work is also fundamentally different from a prior reported sub-Hz integrated Brillouin laser¹⁶, where the linewidth in that work was limited by several factors. First, the relatively small mode volume of the laser cavity set a limit on the ST fundamental linewidth due to limited first order Stokes power and number of photons. Second, the onset of second order Stokes lasing clamps the number of photons in the first order Stokes and injects phase noise back into the first order, driving up the fundamental linewidth in the first order²³. Third, the single longitudinal mode cavity of the smaller cavity¹⁶ sampled the Brillouin gain with a single FSR resonance, limiting the cavity length, mode volume, first order Stokes output power, and fundamental linewidth. The current coil Brillouin laser results eliminates these bottlenecks by enabling an increasing length (hence mode volume) cavity without the onset of multimode lasing and enabling the fundamental linewidth to be continued to drive down lower without noise being fed back from the onset of second order Stokes, and without the onset of

multimode lasing. This results allows the linewidth to scale lower and output power to scale higher due to the Brillouin nonlinear gain producing a single mode laser in a multimode cavity in an integrated waveguide laser.

3) *The 6000 nm by 80 nm silicon nitride waveguide is a multi-modal waveguide. How do the authors prevent the excitation of high-order optical transversal modes? The authors should also add experiment data to the show the optical transmission spectral response of such a 4-meter-long coil waveguide (open status) over a wide wavelength range in optical C and L bands. The authors should provide OFDR loss measurement to cross-prove the loss retrieved from the resonance linewidth measurement.*

Author response: We agree with the reviewer that this waveguide is a multi-mode waveguide. The spectral scanning of the resonator as shown in Supplementary Fig. S3a, shows the *quasi-single-mode* coil resonator, which is achieved by the combination of two techniques: (1) the coil resonator center S bend is a modal filter that makes the higher-order modes (not the TE₀ and TE₁ modes) very lossy and un-supported, and (2) the bus-resonator coupling is optimized for TE₀ so that other modes such as TE₁ mode is either very weakly coupled or overly coupled. We have added references to our previous work that have extensive discussion on this aspect. We think these efforts address the reviewer's concerns without the need for further experimental data.

Revised Text:

- The spectral response in Supplementary Fig. S3a shows the TE₀ resonance only. Although the high-aspect-ratio waveguide is a multimode waveguide, the coil resonator center S bend acts as a modal filter that is lossy for higher-order modes and low loss for the TE₀ and TE₁ modes²⁶. Additionally, the bus-resonator coupling is designed to couple only the TE₀ mode in C and L bands to achieve a quasi-single-mode resonator^{17,27}.

4) *The black elliptic circle in Figure 2 should cover the actual zoomed-in range. It looks confusing.*

Author response: We replaced the elliptic circle with a line bar that clearly indicates the actual zoomed-in range in Fig. 2.

5) *It seems that the laser can only be tuned to several sparse wavelengths, limited by the Vernier effect. Is it correct? Can this issue be overcome? If so, please add data to show laser tuning with denser wavelength spacing.*

Author response: The reviewer's understanding is correct as illustrated in Fig. 4a and 4b. We discuss in the main text that "This translates to a Vernier effect where the SBS phase matching wavelength points every ~6 nm near 1570 nm, as expressed by $\delta\lambda = \lambda v_{FSR}/\Omega_B$ ". In order to reduce the Vernier period in wavelength, the most straightforward approach is to reduce the FSR by increasing the coil resonator length,

which comes at an expense of increasing the laser threshold as it increases with the coil resonator length. This is the subject of future work.

Revised Text:

- In order to reduce the Vernier period in wavelength, the most straightforward approach is to reduce the FSR by increasing the coil resonator length, which comes at an expense of increasing the laser threshold as it increases with the coil resonator length. This is the subject of future work.

6) *The latest work on a high-power narrow-linewidth integrated laser should be included in Figure 5 to ensure better literature coverage.*

Author response: We were not sure which work this comment was referring to, but based on a literature search for the most recent results, we know of the following work - Franken, C. A. A., Cheng, R., Powell, K., Kyriazidis, G., Rosborough, V., Musolf, J., Shah, M., Barton, D. R., III, Hills, G., Johansson, L., Boller, K.-J. and Lončar, M. (2025). "High-power and narrow-linewidth laser on thin-film lithium niobate enabled by photonic wire bonding." *APL Photonics* 10(2): 026107, which reports "At higher currents, the laser produces a high maximum on-chip power of 76.2 mW while maintaining 51 dB side mode suppression. The laser frequency stability over short timescales shows an ultra-narrow intrinsic linewidth of 550 Hz."

We have added this work to the plot in Figure 5.

7) *Please correct typos in "***within an large mode volume***" (Line 23), "***modes withing**" (Line 115), "*** silica clad" (Line 179), etc.*

Author response: We have corrected the typos listed above and carefully checked the manuscript for other typos.

8) *Will the suppression of 2nd Stokes Brillouin lasing help improve the laser performance? Can you estimate it or provide any comments?*

Author response: We have briefly discussed the approach of 2nd Stokes suppression in the introduction in the main text that "Designs that inhibit the onset of second order Stokes lasing have been employed to decrease the linewidth and increase the single mode output power²⁰⁻²². However, in these designs the first-order Stokes (S1) laser emission grows modestly— scaling with the square-root of the pump power— limiting the output power and the linewidth^{22,23}." Although the suppression of 2nd Stokes allow S1 to further increase its output power and reduce its fundamental linewidth, the power increase and linewidth decrease saturates and is not linear with the pump power and becomes inefficient at high pump power. This is why we intentionally use the 4-meter-coil large-mode resonator to have a Brillouin laser with a high threshold to achieve high output S1 power without suppression of 2nd Stokes, allowing the output power and number of photons in S1 to continue to increase.

Reviewer #2

In this paper, the authors demonstrated an integrated stimulated Brillouin laser with 31 mHz fundamental linewidth, 41 mW output power and 73 dB sidemode suppression ratio. This is achieved using a 4-meter coil resonator with an intrinsic Q of 160 million. The experimental results are interesting. However, before I can recommend this work for publication in Nature Communications, several concerns must be addressed.

1. The authors use a 4 meter coil resonator as the laser cavity. The length of the waveguide is comparable to the fiber-based system. How does the performance of the demonstrated laser compare to the state-of-the-art Brillouin fiber lasers in terms of the output power, fundamental linewidth, and the thermal stability? A detailed comparison would provide valuable context for assessing the advantages and limitations of this approach.

Author response: The short-term thermal stability at frequency offset from 1 kHz to 10 kHz is mostly dominated by the resonator-intrinsic TRN that inversely scales with the resonator mode volume for both waveguide-based and fiber-based Brillouin lasers, which is extensively discussed throughout the manuscript. The long-term thermal stability is very much subject to environmental temperature drift and the device thermal isolation design in the packaging engineering, which is not the scope of this work. All fiber-based systems are subject to environmental disturbances and fluctuations and noise sources that are not present in an on-chip waveguide device. Very complicated active stabilization, usually a combination of temperature and bulk PZT actuators, are employed in fiber based systems, including fiber Brillouin lasers. This is not needed with our device. For brief comparison between the fiber Brillouin lasers and the integrated coil resonator Brillouin lasers on the fundamental linewidth and output power, we have added a section in the discussion.

Revised Text:

- Notably, although the fiber based cavity can have similar or larger optical mode volume than the integrated waveguide coil resonators, the single-frequency fiber Brillouin lasers based on the 2-meter-long and 11-meter-long fiber cavities^{44,45} are reported to have measured fundamental linewidths of 7 Hz and 150 Hz, respectively, and optical output powers of 9 mW and 0.19 mW, respectively. Such fiber-based lasers are subject to environment driven fluctuations and noise sources that require complex isolation and thermal and piezoelectric feedback control.

2. If I understand correctly, achieving a high sidemode suppression ratio (SMSR) requires precise alignment of one optical resonance with the peak of the Brillouin gain profile while ensuring that the remaining three optical resonances within the Brillouin gain bandwidth fall in regions where the gain is significantly lower. This requirement would impose significant challenges as the FSR of the cavity reduces. Furthermore, as the cavity length increases, the coil cavity may become more susceptible to thermal fluctuations. Given these considerations, the authors should provide a more detailed discussions of the advantages and limitations of using a longer coil length when making claims about the scalability of this approach.

Author response: We thank the reviewer for this comment. The lasing in this design (and non-coil SBS designs as well) is very robust as the laser will pick the overlap between the highest gain and its overlapping resonator mode. This selection is stronger than the width of a cavity resonance since the winning phonon mode overcomes competition with other FSR modes as well as spontaneous modes as discussed in the manuscript. As the cavity FSR reduces (due to cavity length increase or temperature change), the selectivity will not diminish and susceptibility to thermal fluctuations will only manifest as wavelength shifting as with most lasers. The selection process is so strong (as also described in the simulation in the Supplemental as well as other publications) that the linewidth narrows by many orders of magnitude of a single cavity resonance itself where even spontaneous modes under a single resonance cannot compete. This behavior, fundamental to the Brillouin process, is what provides the ability to continue to scale the size of the coil cavity. This is the subject of current and future work. We have provided strong evidence of this in the current manuscript and in the modeling and simulation. This is why the SMSR and operation are so robust to variations in exact location of cavity resonance relative to the Brillouin gain profile (e.g. temperature, fabrication variation), for example we can Vernier tune the laser. We have included detailed simulation and modeling in the Supplemental section and a brief description in the text that addresses the highly robust lasing mechanism.

Revised Text:

In the case of lasing mode selection and robustness, the laser will select the overlap between the highest Brillouin gain frequency and its nearest overlapping resonator mode. Due to the large Brillouin gain and wide gain bandwidth provided by non-guided phonon modes¹⁶, the mode selection and SMSR are robust to exact positioning of the cavity resonance relative to the gain. The combination of strong Brillouin nonlinearity, high integrated resonator Q, and short phonon lifetime of the non-guided phonons enable large linewidth narrowing of the main lasing mode by inhibiting spontaneous emission within the coil resonance. This fundamental property of the Brillouin nonlinearity will allow this design to continue to scale to longer cavities with smaller FSR. With temperature changes and fluctuations the Brillouin gain center frequency tuning and change in the FSR results in wavelength tuning or Vernier tuning. Such changes may be desirable as is the case with a tunable Brillouin laser or would require temperature stabilization of the chip or athermal resonator designs.

3. The authors stated that the SBS laser can be tuned every 48 MHz on the coil resonator FSR grid, however, they also mentioned that the SBS phase matching condition is only satisfied every 6 nm. Also, in Fig. 4b, SBL is only shown every 6 nm. These two claims seem contradictory. Could the authors clarify this apparent discrepancy? Furthermore, the description of the Vernier tuning scheme is not entirely clear and would be nice if the authors can provide a more detailed explanation.

Author response: We have revised the text to add clarity on the continuous tuning and the Vernier effect. We have noted that the continuous tuning and one FSR tuning are not specifically demonstrated in this paper (prior work or next work) and have adjusted the Vernier figure to incorporate this discussion, see below.

Revised Text:

- We additionally demonstrate that this approach can be used to realize a broadly tunable Vernier tunable SBS laser. The SBS laser can be fine-tuned around each lasing resonance (less than an FSR) by adjusting the coil resonator temperature as has been previously shown⁴⁹. By tuning the pump to a neighboring resonance (one FSR away), the laser output can be tuned to a neighboring 48 MHz resonance as long as the wavelength dependent Brillouin frequency shift remains small compared to the FSR. This condition can hold for a pump tuning over a small number of FSRs. When the pump laser is tuned by more than a small number of resonances, there is a pump wavelength dependent Brillouin gain frequency shift that occurs, causing the gain to overlap with a resonance that is a different integer number of FSRs away from the pump, creating a Vernier condition. This occurs since the dependence of the Brillouin frequency shift on the pump mode wavelength is $\Omega_B \propto 1/\lambda$, and since the coil resonator FSR is smaller than the Brillouin gain bandwidth (in this case ~50 MHz FSR compared to 250 MHz), the phase matching condition takes on a Vernier relation, in this case every ~6 nm. This is illustrated in Fig. 4a where the SBS phase matching condition $\Omega_B(\lambda) = N v_{FSR}$ is satisfied at one wavelength point (λ) and for a second wavelength point ($\lambda + \delta\lambda$) the SBS-phase-matched wavelength occurs at $\Omega_B(\lambda + \delta\lambda) = (N-1)v_{FSR}$ is $\delta\lambda = \lambda v_{FSR} / \Omega_B$. For this designs, this Vernier relation is ~6 nm around 1570 nm operation. This behavior is enabled by the small FSR (due to the long coil length) compared to the 250 MHz gain bandwidth, a condition which is not possible for waveguide ring resonators with FSRs of a few GHz.

Fig. 4. Wide-wavelength-range Vernier tuning in a coil SBS laser in C and L band. a, The SBS phase matching condition is satisfied for a Brillouin shift of approximately 10.6 GHz. The Brillouin gain bandwidth is about 250 MHz wide due to the non-confined phonons (bulk-like). Thermal tuning of the cavity to less than an FSR provides fine tuning of the SBS output (first order Stokes S1) as demonstrated in prior work. b, The Brillouin output can be medium tuned by adjusting the pump over a small number of FSR which will track the Brillouin gain (not demonstrated in this work). For wide tuning, the Vernier effect is demonstrated (lower illustration), where the Brillouin shift Ω_B is linearly proportional to the optical frequency and changes by more than an FSR when the pump wavelength changes by ~6 nm around 1570 nm. c, S1 emission recorded on the OSA reaches a local minimal threshold every ~6 nm in C and L band. OSA, optical spectrum analyzer. Ω_B , Brillouin shift frequency. FSR, Free spectral range. SBS, Stimulated Brillouin Scattering. OSA, Optical spectrum analyzer.

How does the optical spectrum would look like if the optical resonance does not perfectly locate at the highest point of the Brillouin gain profile? It would be helpful to show the optical spectrum at wavelengths beyond those presented in Fig. 4b to provide a more comprehensive understanding of the tuning behavior.

Author response: This is an excellent observation. We are studying this behavior now, and it is outside the scope of the current work; we plan to report a more comprehensive study including this in the future. The SBS laser's behavior regarding lasing threshold, output power and fundamental linewidth has been investigated and reported in literature in conventional small-size waveguide ring resonators, where for off resonance, the threshold increases as well as the linewidth enhancement factor. The results that we report in Fig. 4 is that peak SBS phase matching can be realized at multiple wavelengths rather than at only one wavelength in conventional waveguide ring resonators with a Vernier behavior. At wavelengths where the optical mode does not locate on the peak of the Brillouin gain profile, we have observed an increase in threshold. This observation is not unexpected. For this paper, we have reported only the OSA traces at wavelengths where the threshold reaches a local minimum, and we have included this new finding in this manuscript.

Revised Text:

- It has been reported in literature that in between the optimally phase-matched wavelengths, where the lasing mode is not on the peak of the Brillouin gain, the SBS laser threshold increases and fundamental linewidth increases due to the increase in the linewidth enhancement factor^{17,34}.

4. In Fig. 3, the authors present the frequency noise measurement of the SBLs and the free-running pump. However, the SBL is measured when the pump is locked to the cavity through the PDH loop. It would be helpful if the authors can also provide the frequency noise spectrum of the locked-pump. This additional data would provide a clearer understanding of how the SBL demonstrated here can suppress the frequency noise of the pump laser through the SBS process.

Author response: We have included the frequency noise of locked pump laser in Fig. 3, and discussion on the suppression of the pump laser noise. As expected, the low to mid frequency noise, within the bandwidth of the servo lock, is brought down to the TRN limit. This noise profile is further reduced by several orders of magnitude by the Brillouin process for frequencies above several hundred kHz as shown in the revise plot in Fig. 3

Revised Text:

- The low to mid frequency pump noise is reduced towards the TRN floor when PDH locking the pump to the coil resonator (green trace in Fig. 3). The Brillouin process further reduces the frequency noise by several orders of magnitude above several hundred kHz where the pump-

transferred frequency noise gets suppressed with the increasing pump power³¹. Above 10 MHz the measured frequency noise is dominated by photodetector noise (dashed light blue trace in Fig. 3).

5. In the frequency noise measurements, the white noise floor of the SBL is not clearly visible, making the claim of the 31 mHz fundamental linewidth less convincing. Could the authors provide more comprehensive characterizations of the frequency noise of the SBL? Additionally, the authors employ a 200-m fiber MZI as the optical frequency discriminator, would a shorter fiber spool help to resolve the noise floor better?

Author response: To the best of our knowledge, the most comprehensive characterization of laser fundamental linewidth is by measuring the white frequency noise floor using an MZI optical frequency discriminator with the correct length. A shorter length will move the MZI FSR fringes out but will limit the slope discrimination resolution. We understand the reviewer's concern that in Fig. 4 the white frequency noise floor has the servo bump width, the MZI FSR fringes, and the PD noise floor creating a bound. We describe and reference the way we access this level in our previous work [22] and in the Methods section by analyzing the frequency minimum. Given the measurement limits, we believe that the claim of 31 mHz fundamental linewidth represents the upper bound since true white frequency noise floor is below the measured value due to the above factors. Therefore, we think that our claim is valid. To address the reviewer's concerns, we have revised the related text.

Revised Text:

- Given the existence of the excess noise from the OFD frequency noise measurement (e.g. servo bandwidth and quality factor, MZI FSR, and PD noise floor) and residual pump-transferred noise, the claim of the 31 mHz fundamental linewidth represents an upper bound of the fundamental linewidth. The OFD frequency noise measurement is optimized to minimize the measurement noise as described in detail in the previous work²² and the methods section.

Additionally, the authors employ a 200-m fiber MZI as the optical frequency discriminator, would a shorter fiber spool help to resolve the noise floor better?

Author response: No. A shorter fiber spool can have a higher measurement noise floor from photodetector noise and other noise sources at high frequency offsets as well as a poorer frequency discrimination. A longer than 200 meters fiber spool would not have a lower measurement noise floor from photodetector noise. Interested readers can make estimates based on the model and analysis in our previous work [22] and the Methods section.

Some small comments:

1. What does the purple line indicate in Fig. 5? Please clarify this in the figure legend or the main text.

Author response: We have removed the purple line.

2. *What is the slope efficiency of the tested SBL?*

Author response: The slope efficiency is 36%, which is added in the main text.

3. *What is the relation between the fundamental linewidth of the SBL and the pump power? How do the measured values compared to the theoretical estimation?*

Author response: The relation between the two in Brillouin lasers is well investigated and understood in the literature. The S1 fundamental linewidth reduces with increasing pump power before the S2 emission, and increases after S2 emission, and has a minimum fundamental linewidth, $\Delta\nu_{ST,min}$, which inversely scales with the resonator length. As discussed in the main text that in our previous work the Brillouin laser based on a 11.83-mm-radius ring resonator, the 4-meter-coil resonator Brillouin laser is estimated to have a fundamental linewidth of 13 mHz. Yet, our main claim is still based on the measured 31 mHz.

- “Although we can measure a fundamental linewidth of 31 mHz, the estimated fundamental linewidth in this 4-meter-coil SBS laser operating at the S1 clamping point is lower than the measured value. In previous work^{16,33}, we measured an SBS laser fundamental linewidth of 0.7 Hz at the S1 clamping point in a 11.83-mm-radius ring resonator with a cavity length of 74.3 mm. Here, the 4-meter-coil cavity length is 54 times the ring resonator cavity length, resulting in the fundamental linewidth expected to decrease by 54 times to 13 mHz.”

Reviewer #3

The manuscript titled “Large Mode Volume Integrated Brillouin Lasers for Scalable Ultra-Low Linewidth and High Power” presents a high power, ultra-low linewidth laser based on Brillouin scattering. Over the last decade, Brillouin lasers have emerged as the highly coherent source for applications in Quantum technologies, Soliton combs, microwave signal generation, optical atomic clocks, and optical communications. While ultra-low linewidth Brillouin lasers have been achieved in different platforms, lack of high output power from these lasers necessitate the use of optical amplifiers, which compromises the advantage obtained from its narrow linewidth.

A high power, ultra-narrow linewidth laser with tunable operation is therefore highly desirable. The current manuscript is therefore a major step forward in achieving a source suitable for real-world applications.

The manuscript, however, lacks clear explanation and assumes that the readers will be experts in Brillouin scattering, which is not always the case. I suggest the authors to address the comments below.

- Line 73: “where an external laser is used to generate phonons in a resonator...”.

How does the external laser generate phonons? Typically, the pump signal first gets scattered from thermally generated phonons to create spontaneously generated Brillouin Stokes, which buildup through

cavity feedback to achieve stimulated emission. In that process, beating between the pump and spontaneously generated Stokes create an acoustic wave at the beat frequency.

Author response: Yes the explanation by the reviewer is mostly correct. We have pointed the reader to our original 2019 publication in Nature Photonics which explains this process in great detail silicon nitride structure where there is no guided phonon structure (leading to short phonon lifetime) and ultra-low loss waveguides in an ultra-high Q resonator (the other condition for Brillouin noise reduction). The reviewer is correct that below threshold the phonons are generated by spontaneous Stokes in all directions which a portion scatter back in the waveguide and interfere with the pump to start forming moving bulk non-guided gratings. With cavity buildup and increased pump, and other dynamics of the grating formation process (which we go into now with a modeling discussion in the updated Supplementary section), a long, high contrast primary gratings forms which takes the pump photons from the other modes, including spontaneous modes, to form the primary lasing mode with high SMSR. Due to the limited space, we did not go through the complete background of the Brillouin process in this type of structure. We did spend a bit of text and a full figure (Fig. 1) explaining the presence of spontaneous Brillouin gratings (Fig. 1d) below threshold, and the onset of lasing at the primary frequency with the dominant grating (Fig. 1e).

- Line 103-107: “provides extreme down selection of both the pump laser photons and the vacuum driven spontaneous emission modes. The Brillouin nonlinear dynamics is so selective that vacuum driven spontaneous modes within the primary resonance and all other cavity resonances that overlap the Brillouin gain are excluded from utilizing pump photons for scattering except for the very select modes”.

- What do authors mean by “extreme down selection of...” and “Brillouin nonlinear dynamic is so selective”. Please clarify what is extreme down selection and how Brillouin dynamics is so selective. If the Brillouin dynamics is so selective then why it does not allow similar linewidth narrowing in other platforms. It must have the same selectivity in other platforms/devices as well. I suggest authors to clearly explain what is the reason for the selective nature of the Brillouin scattering as it is not explained well.

Author response: Yes, we agree, this is the fundamental behavior of Brillouin. The manifestation of this behavior in an integrated large mode volume coil resonator is what is novel and new. We have described this process in this manuscript and a prior publication in a way that is different than what has been described before (see ref. 23 and the new material in the supplemental section in this manuscript). The extreme down selection refers to the buildup of the first primary grating and the highly nonlinear feedback of Brillouin (stronger than FWM), to continue to “steal” the pump photons as the grating builds up, which also builds up the contrast of the grating, which then takes more pump photons in a continuing cycle from the other resonator modes and spontaneous modes. We have shown this in the below threshold and above threshold spectral output curves. Brillouin dynamics are selective from the process of grating buildup becoming more selective, the effective gain from the coil-length multiplied by the resonator Q,

Line 111: The statement “This process is defined by the strength and bandwidth selectivity of the dominant Brillouin grating that scatters the output Stokes light.” may give a non-expert an impression that Brillouin gain bandwidth and selective nature of the Brillouin scattering are the same.

Author response: Yes we agree this language is confusing, thank you for catching this. We have made the following update.

Revised Text:

- This process is defined by the strength and frequency selectivity of the dominant Brillouin grating that scatters the output Stokes light.

The authors should clearly explain the distinction between the selective nature of the Brillouin scattering and bandwidth of the Brillouin gain profile. Otherwise, it may lead to confusion because the Brillouin gain bandwidth here is 250 MHz, which is 8 times larger than the Brillouin gain bandwidth (~ 30 MHz) in an optical fiber, so why such a linewidth narrowing is not achieve in fiber-based Brillouin lasers. The highly selective nature of SBS alone is not responsible for the narrow linewidth of the Brillouin laser, Q of the cavity resonance has a role to play here.

Author response: We agree, the broadened gain bandwidth due to our non-guided phonons is not responsible for the highly selective nature. It is the extremely large gain and feedback of the Brillouin process coupled with the ability to make very stable long Brillouin gain and resonator cavities on chip, and large single pass cavities with extremely high Q (leading to very high effective length via the coil length and Q over which the Brillouin interaction can take place) which is difficult to make in fiber (and stable). Fiber Brillouin lasers do employ linewidth narrowing based on the same physics (although the phonon modes are quite different in the planar and cylindrical structures), and we think this is more apparent in the waveguide structure. The fiber based lasers goes to great lengths to ensure single mode operation through additional filtering, so it is likely this effect has not been carefully studied, as far as we are aware. We also are not aware of this discussion being previously published. It is the extreme condition that we show with the Brillouin gain bandwidth spanning 5 x FSR that highlights this result. We have also now included a detailed theoretical analysis of this behavior in our waveguide in the updated Supplementary section, which we will broaden out to be a subject of a future publication.

Revised Text:

- See new Section in the SI Supplementary Note 2: SBS laser coupled mode model and single-mode lasing.
- Discussion: “As a note to Brillouin lasing in comparison to fiber implementations. The broadened gain bandwidth due to our non-guided phonons is not primarily responsible for the highly selective nature. It is the extremely large gain and feedback of the Brillouin process coupled with the ability to make very stable long Brillouin gain and resonator cavities on chip, and large single pass cavities with extremely high Q . This leads to very high effective length via the coil length and Q over which the Brillouin interaction can take place which is difficult to do stably in fiber. Fiber Brillouin lasers

employ linewidth narrowing based on the same physics, although the phonon modes are quite different in the planar and cylindrical structures, yet as far as we are aware this description of the process has not been described before. There is also the possibility that the low loss, high Q, and large effective length of the integrated coil resonators leverages the long photon coherence to phase lock the bulk phonon modes all along the resonator, creating a super grating over the whole structure that is highly frequency selective. These topics are the subject of current and future work.”

Authors have repeatedly used terms like “Physics of Brillouin laser”, “highly selective nature of Brillouin lasing”, “Brillouin lasing physics...” throughout the text. All of these refer to the same phenomenon yet that phenomenon has not been clearly explained. It would be good to explain it in the beginning so that it clarifies why Brillouin scattering allows selective amplification.

I will avoid use of superlatives such as extreme.

Author response: We have removed all superlatives. We have inserted a brief discussion of the Brillouin phenomenon at the beginning of the manuscript. We have been brief in the revision and referred to our prior publications that cover this topic in more detail (*Gundavarpu et al., Nature Photonics, 2019, Behunin, R. O. et al., Physical Review, 2019*).

Revised Text:

- “In the present work, the resonator is the nonlinear laser cavity itself, where Brillouin physics^{1,2} inside the cavity provides down selection of both the pump laser photons and the vacuum driven spontaneous emission modes. The Brillouin nonlinear process is so selective that vacuum driven spontaneous modes within the primary resonance and all other cavity resonances that overlap the Brillouin gain are excluded from utilizing pump photons for scattering except for the very select modes.”
- A critical point is that the Brillouin laser cavity can be implemented in a complex waveguide geometry such as the coil resonator owing to the physics of this type of Brillouin laser, where the photon-phonon scattering interaction occurs over the length of the resonator without using phonon guiding^{16,23}. Below threshold the Brillouin process is dominated by filtered spontaneous emission, where pump photons are scattered randomly in a manner that preferentially populates the Stokes modes that lie within the gain bandwidth (see Fig. 1). As the pump power is increased, the beat note of pump light with the spontaneous emission form stress-optic gratings that enhance the backscattering of frequency shifted Stokes photons at the cavity resonances. The Stokes mode with the largest emitted power, forms the deepest stress-optic grating, further enhancing the emitted power. As a result of this form of feedback, the Stokes mode with the largest gain (i.e., emitted power) reaches threshold first, where the SBS gain is precisely balanced by the loss, and the intracavity pump power is clamped. As a consequence, the emission of the subthreshold Stokes modes is fixed and only the power of the lasing mode increases as the supplied power increases.

- Line 131: “A critical point is that the Brillouin laser cavity can be implemented in a complex waveguide geometry such as the coil resonator owing to the physics of this type of Brillouin laser, where the photon-phonon scattering interaction occurs over the length of the resonator without using phonon guiding.”

What is the relevance of phonon guiding here?

Author response: The lack of phonon guiding in our structure leads to the ability to generate resonant Brillouin gain in a fairly arbitrary shaped structure, for example a coil resonator. This is not possible, or has never been done due to the difficulty, when one has to make the optical waveguide at the same time a phonon guiding structure that also has to be phase matched (photon and phonon group velocities). This lets us make a coil Brillouin resonator cavity and also makes it easier to make the Brillouin laser work at almost any wavelength in the visible to SWIR. Additionally, the lack of guiding generates a broader Brillouin gain. This is discussed in detail in our Nature Photonics 2019 paper.

- Line 134: “The absence of phonon waveguiding leads to a broad Brillouin gain spectrum of approximately 250 MHz at 1550 nm.....”

How does the absence of phonon waveguiding leads to a broad Brillouin gain spectrum? Is phonon waveguiding present in other platforms where the linewidth is narrow? The Brillouin gain bandwidth depends on the phonon lifetime in a device. If the authors want to convey that the absence of phonon waveguiding results in a smaller phonon lifetime then please elaborate this point.

Author response: The lack of phonon guiding in our structure leads to a broader Brillouin gain than in waveguide that guide and phase match both phonons and photons. This is discussed in detail in our Nature Photonics 2019 paper. Yes all other papers have employed creating guided phonon structures, some are starting to employ our technique. Since this is described in detail in our prior publication we would prefer not to do so here again. Also, this is why we can make these lasers work at almost all wavelengths from Visible to SWIR is since we do not have to satisfy both low loss photon and low loss phonon guiding at the same time.

- Line 174: “In this mode of operation, it is desirable to increase the optical mode volume by increasing the cavity length...”

This statement is counterintuitive as the Brillouin lasing threshold in a microresonator is proportional to the pump mode volume. So, increasing the pump mode volume will result in increased threshold. The Brillouin threshold does not depend on the cavity Q alone but also depends directly to the mode volume. Since the Brillouin gain varies with the interaction length L according to $gBIP$, where gB and IP refer to the Brillouin gain coefficient and pump intensity respectively, increasing the cavity length makes more sense.

Author response: Yes, we have made this point in the existing paper, that we purposely increase the S1 threshold in order to drive out the S2 threshold, so that we can continued to drive up the S1 photon population before the onset of S2 without reaching clamping of S1 while trying to generate high output power and drive down the fundamental linewidth. This is a large part of the basis of this work.

Other comments:

Please go through the text carefully to avoid repetitions and sentence construction. Here are some examples.

Lines 68-69: “to selectively reduce the existing spontaneous in the single mode laser for the former, or drive.....”. In this sentence “existing spontaneous” should be “existing spontaneous emission”

Author response: Thank you, yes we have addressed this issue in the revised manuscript.

Line 74: “nonlinear feedback between input and output photons and phonons”. What does “input and output photons and phonons” refer to in the above sentence? What are “input and output phonons” here?

Author response: Input photons are pump, output photons are S1 and phonons are the intermediary. We have clarified this in the text.

Line 114: “ “steals” all of the pump photons to the drive the single lasing mode....”. This sentence should be rephrased to “...pump photons to drive the single mode lasing...”

Author response: Corrected.

Line 115: Check Spelling of “within”

Author response: We are not sure where this is referring to.

Line 206: Instead of using “dropped by a fiber circulator” it is better to use “collected using a fiber circulator”.

Author response: Fixed.

Line 212: “all increases in pump energy are directed...”

Author response: We have replaced with the following.

Revised Text:

Above threshold, all photons resulting from increased pump are scattered to the primary lasing mode, enabling this large contrast between the power in the lasing mode and the side modes.

Line 277: “...Brillouin shift frequency’s dependence on the pump mode wavelength...” may

Author response:

be rephrased to "...dependence of the Brillouin frequency shift on the pump mode wavelength..."

Author response: Fixed, see above response to Reviewer #1.

Line 278: Rephrase "to achieved". It would be better to write the actual expression for Ω_B .

Author response:

Author response: Fixed, see above response to Reviewer #1.

Reviewer #4

K. Liu et al. demonstrate an on-chip single-mode laser with an impressive narrow linewidth of 34mHz and output power of over 40mW, tunable over 22nm. The performance is made possible due to the low-loss 4m-long Si₃N₄ integrated waveguides and represent a milestone in Brillouin lasers. The experimental results are well presented and thoroughly discussed. The authors use a precise method to measure the very narrow linewidth of the laser. In general, I recommend publication in Nature Communications because this result is important for a number of fields (not only nonlinear optics and Brillouin scattering). However, I have some comments that should be addressed beforehand.

- The same authors published already sub-Hertz Brillouin lasers (Nature Photonics 2018) and very recently a Brillouin laser with below 100mHz linewidth and 11 mW output power (Optics Letters 2024). So one criticism is: what is the fundamental breakthrough here, why is this particular laser configuration so good, is it based on the ability to fabricate so uniform, low-loss and long integrated waveguides to achieve such stable configuration?

Author response: We thank the reviewer for this question. The fundamental breakthrough is first time Brillouin lasing in a large mode volume integrated coil resonator. This has never been achieved before. Also the ability to demonstrate single mode lasing in a long cavity with multiple FSRs overlapping the Brillouin gain has never been demonstrated. Also, the mitigation of onset of S2 lasing threshold by moving to a large volume resonator to enable high power at the same time as low fundamental linewidth has never been demonstrated.

- The authors write that the laser should have a linewidth of 13 mHz but pump-transferred noise prevents the measurement of a linewidth lower than 31 mHz (one paper of the same group of authors is cited). Could the authors give some more details in this manuscript and for this laser configuration about it, too?

Author response: Thank you for this important question. We have replaced the text with the following to clarify and mention this is the subject of future work.

Revised Text:

Here, the 4-meter-coil cavity length is 54 times the ring resonator cavity length, resulting in the fundamental linewidth expected to decrease by 54 times to 13 mHz. The ability to measure low magnitude frequency noise and fundamental linewidth at 0.1 MHz and above is limited by the frequency noise measurement system. The ability to reliably make these measurements are a subject of future work.

- About the single-mode operation: it is impressive to see that the strict phase matching condition apparently allows to enhance only one mode even if 5 modes lay within the Brillouin resonance. There is a model given in the supplementary but as far as I see, no simulations are presented.

Author response: Thank you for the recognition of achievement, it is very much appreciated. We have updated the SI (Note 2) and mentions in main text to include the full model and simulation now. See revised materials.